# Retrospective Comparison of the Anesthetic Effects of Tiletamine–Zolazepam with Dexmedetomidine and Ketamine with Dexmedetomidine in Captive Formosan Serow (*Capricornis swinhoei*)

**DOI:** 10.3390/ani14101413

**Published:** 2024-05-08

**Authors:** Li-Jen Chang, Hsin-Yi Weng, Chen-Yeh Lien, Kuan-Sheng Chen

**Affiliations:** 1Department of Veterinary Medicine, College of Veterinary Medicine, National Chung Hsing University, Taichung 40227, Taiwan; ljchang@vt.edu; 2Department of Small Animal Clinical Science, Virginia-Maryland College of Veterinary Medicine, Blacksburg, VA 24061, USA; 3Department of Comparative Pathology, College of Veterinary Medicine, Purdue University, West Lafayette, IN 47907, USA; 4Animal Medical Center, Taipei Zoo, Taipei 11656, Taiwan

**Keywords:** Formosan serow (*Capricornis swinhoei*), tiletamine–zolazepam, dexmedetomidine, induction, recovery

## Abstract

**Simple Summary:**

The Formosan serow is a wild goat found only in the mountains of Taiwan. In this cross-over study, two different drug combinations were used to anesthetize five adult Formosan serows in a zoo environment, namely dexmedetomidine–tiletamine–zolazepam and dexmedetomidine–ketamine. Each animal received each of the drug combinations for general anesthesia at least once, with an interval of at least 12 months between anesthesia administrations. While dexmedetomidine–tiletamine–zolazepam induced anesthesia faster than ketamine–dexmedetomidine, these animals experienced problems such as paddling, prolonged recovery, and ataxia. The animals in the dexmedetomidine–tiletamine–zolazepam group had lower breathing rates and body temperatures than those in the dexmedetomidine–ketamine group. In summary, the dexmedetomidine–tiletamine–zolazepam combination anesthetized the Formosan serows rapidly but recovery was more eventful.

**Abstract:**

Formosan serows are endemic to the mountainous regions of Taiwan. This crossover study aimed to assess and compare the anesthetic induction and recovery using either dexmedetomidine–tiletamine–zolazepam (DZ) or dexmedetomidine–ketamine (DK) by intramuscular injection from a blow-dart in a zoo environment. Ten anesthetic procedures were performed with five adult Formosan serows. Each participant was anesthetized with both combinations at least once with a minimal 12-month washout. The average dosages were 22.6 ± 8.3 µg/kg and 35.8 ± 2.5 µg/kg for dexmedetomidine and 185.6 ± 123.6 and 357.8 ± 25.2 µg/kg for atipamezole for the DZ and DK groups, respectively. The doses of tiletamine–zolazepam and ketamine were 2.1 ± 0.25 mg/kg and 3.6 ± 0.3 mg/kg, respectively, in the DZ and DK groups. All participants were induced within 10 min (median: 8 min for both groups), except one serow in the DK group with an induction time of 22 min. Serows in the DZ group had a lower respiratory rate (*p* = 0.016) and lower rectal temperature (*p* = 0.008) than those in the DK group. The quality of recovery was poor for DZ because of paddling, prolonged recovery, and ataxia after antagonism of dexmedetomidine with atipamezole. The induction of anesthesia with dexmedetomidine–tiletamine–zolazepam was uneventful and rapid. However, recovery from this combination was not smooth.

## 1. Introduction

Formosan serows (*Capricornis swinhoei*) are endemic to the mountainous areas of Taiwan. The species is listed as of least concern by the International Union for Conservation of Nature (IUCN). However, it is a “precious and rare species” protected under the Wildlife Conservation Act of Taiwan because of habitat loss and heavy hunting [1,2].

Anesthetizing species in a zoo environment is a challenge, so the anesthesia injectable via a blow dart may be useful for practitioners to perform diagnostic and surgical procedures. However, few studies have assessed the anesthetic effects of different combinations of anesthetics for Formosan serow. The anesthetic effects of ketamine-based combinations have been previously assessed in this species, but the options for anesthesia remain limited [3].

Tiletamine–zolazepam is a readily available commercial veterinary anesthetic agent in most countries and has been used as a part of safe anesthesia protocol in various animals, including small ruminants [4], camelids [5], and wildlife [6]. Tiletamine and zolazepam are combined in a 1:1 ratio by weight of the base and labeled as Telazol^®^ (Zoetis, Parispanny, NJ, USA) in North America and Zoletil^®^ (Zoletil^®^; Virbac, Taipei, Taiwan) in other countries. Zoletil 50^®^ is the most commonly used commercially available product, and it contains 125 mg of tiletamine and 125 mg of lyophilized zolazepam powder in a vial. This combination of drugs has a wide margin of safety, leads to better muscle relaxation, and has more profound analgesic effects than ketamine alone [6]. Lyophilized tiletamine–zolazepam powder should be reconstituted with sterile water but other injectable drugs, such as dexmedetomidine and butorphanol have been used to reconstitute the powder instead. One well-known reconstitution protocol is tiletamine–zolazepam–butorphanol–dexmedetomidine (TTDex), which has been used extensively in high-volume, high-quality shelters [7,8]. Another potential advantage of tiletamine–zolazepam combinations over ketamine-based anesthetic protocols in wildlife species is the relatively lower volume for intramuscular injection [9], which is of paramount concern when an anesthetic must be delivered via a blow dart [10].

Alpha-2 adrenergic receptor agonists, such as xylazine or dexmedetomidine, have been used extensively to provide effective sedation and analgesia in ruminants. Dexmedetomidine is a racemic enantiomer with high alpha-2 selectivity. It reportedly induces stronger sedation, muscle relaxation, and analgesia than other alpha-2 adrenergic receptor agonists in several domestic species, including dogs, cats, and sheep [11,12,13]. However, its adverse cardiovascular and respiratory effects remain a concern. It has been reported that goats are more sensitive to xylazine than sheep and camelids, and they are more prone to suffering from adverse effects, such as a decrease in cardiac output, respiratory distress, and hypoxemia, and require a lower dose to achieve a similar degree of sedation [14,15,16]. However, information regarding the use of dexmedetomidine in captive small ruminants, particularly wild goats, is scarce. Atipamezole is a high-selectivity alpha-2 adrenoceptor antagonist, and it is currently preferred over other antagonists because of the increasing use of highly specific agonists, including medetomidine and dexmedetomidine [17]. It has also been used successfully for the reversal of the effects of xylazine and medetomidine-based protocols in various sheep species [18]. The risks of patient excitation, loss of analgesia, and adverse cardiovascular effects, however, can be observed after the reversal of the effects of alpha-2 agonists [17,19].

Ketamine is a dissociative anesthetic used extensively in animals for immobilization and anesthesia and has been successfully used for Formosan serow anesthesia [3,6]. Ketamine can be administered intramuscularly to anesthetize animals that are not easily restrained for intravenous administration of drugs; therefore, it has been used extensively in zoos and on wildlife species [20]. However, ketamine is notorious for muscle rigidity after administration; therefore, the concurrent administration of muscle relaxants such as alpha-2 agonists or benzodiazepines is recommended [21]. Abnormal behaviors, including transient vocalization, ataxia, hyperreflexivity, sensitivity to touch, and increased motor activity, have been reported in animals recovering from ketamine-induced anesthesia. These reactions are usually temporary and resolved spontaneously but remain a concern during recovery from anesthesia as the risk of injury may be increased. The administration of other sedatives or tranquilizers, such as alpha-2 agonists, acepromazine, or benzodiazepines, may decrease the incidence and severity of unsatisfactory adverse effects during recovery [22].

It has been reported that myopathy could lead to an increase in the concentration of muscle-origin serum biochemistry enzymes, including lactate dehydrogenase (LDH), creatinine kinase (CK), aspartate aminotransferase (AST), and alkaline phosphatase (ALKP) [23,24]. Therefore, monitoring the above-mentioned enzymes may help detect capture myopathy in Formosan serows. Nonetheless, measurements of leukocytes, stress hormones, immune functions, and blood glucose concentrations are valuable for detecting stress in free-ranging vertebrates [25].

This study aimed to compare the anesthetic induction and recovery effects of tiletamine–zolazepam combined with dexmedetomidine and ketamine combined with dexmedetomidine, and to assess potential side effects, including the heart rate, respiration, oxygenation, body temperature, anesthesia-associated muscular injury, and stress, of this novel combination in captive Formosan serow. We hypothesized that tiletamine–zolazepam combined with dexmedetomidine would exert anesthetic effects superior to that of ketamine with dexmedetomidine, with minimal side effects in captive Formosan serow.

## 2. Materials and Methods

Five Formosan serows (2 males and 3 females) kept in captivity at the Taipei Zoo, Taipei, Taiwan (24°59′42″ N, 121°35′3″ E), were included in this study. Approval for animal use was not mandated for animals requiring clinical interventions, including annual health examinations. Owner consent was not applicable to this study.

The D (Dexdomitor^®^; Zoetis, Taipei, Taiwan) K (Imaldene 1000^®^; Merial, Taipei, Taiwan) combination was used in five anesthetic protocols from December 2009 to November 2013. The DZ (Zoletil 50^®^; Virbac, Taipei, Taiwan) combination was used in five anesthetic protocols from January to July 2015. Each participant was anesthetized twice, and each protocol was used once, with an interval of at least a year. Anesthesia was administered as part of the routine healthcare of the serows for procedures, including annual health examinations, trauma management, castration, enucleation, and ophthalmologic examinations. The duration of procedure of the cases included in this study was no more than 120 min. The American Society of Anesthesiologists Physical Status (ASA-PS) was used to determine the anesthetic risk. ASA-PS I and II indicate good general health or mild systemic disease, which are associated with lower anesthetic risks [26]. The participants were evaluated and determined to have low anesthetic risks (ASA-PS 1–2), and food and water were withheld for at least 12 h before anesthesia.

The dosage of the anesthetic was calculated based on body weight from visual measurements and previous medical records. The criteria for the determination of the actual dosages were based on the body weight and the temperament, such as stressful, fearful, anxious, aggressive, and so on of each subject. For the DZ combination, the reference dose of dexmedetomidine was 40–60 µg/kg, and that of tiletamine–zolazepam was 2.2–4.2 mg/kg. For the DK combination, the reference dose was 30–50 µg/kg for dexmedetomidine and 3–4 mg/kg for ketamine. The anesthetics were mixed in a 3 mL syringe and administered intramuscularly via a blow dart by two well-trained veterinarians (LJC and CYL) who were not blinded to the combination. Atipamezole (Antiseden^®^; Zoetis, Taipei, Taiwan), the antidote for dexmedetomidine, was administered at the same volume as dexmedetomidine after the procedures, and the animals were returned to their enclosures. The route of atipamezole administration was determined by a clinician based on the clinical signs of the patient during the procedure. Atipamezole was administered intramuscularly and intravenously (the volume was divided in half) to patients who experienced intraoperative complications, such as hyperventilation, hyperthermia, or hypoxemia. For most anesthetic protocols, atipamezole was administered intravenously.

The induction time was defined as the time at which the anesthetics were completely injected intramuscularly until the animal showed lateral recumbency. When induction was complete, the animal was relocated to the animal medical center of Taipei Zoo for intratracheal intubation and intravenous catheterization. The distance between the enclosure and the animal medical center was approximately two kilometers, and transportation took approximately five to ten minutes. Oxygen (10 L/min) was administered via an insufflation tube to the patients during transportation, and their necks were elevated using a towel with the mouth and nostrils downward to prevent aspiration. The serows were intubated with five-to-seven-millimeter (I.D.) endotracheal tubes (Medline, Northfield, IL, USA), depending on body weight. If the first intubation attempt failed, isoflurane (5%) was administered via a mask to facilitate intubation. An intravenous catheter was placed on either side of the cephalic vein with a 20 gauge IV catheter (Terumo, Taipei, Taiwan). Anesthesia was maintained using isoflurane in a rebreathing circuit. The oxygen flow rate was set to 20–40 mL/kg/min. Isoflurane and oxygen were delivered by the Dayex–Ohmeda Excel 210SE^®^ (Ohmeda, Madison, WI, USA). The animals breathed spontaneously throughout the procedure. Blood samples were then collected after placing an IV catheter and stored at 4 °C for further analysis. Alanine transaminase, aspartate transaminase, alkaline phosphatase, creatinine kinase, lactate dehydrogenase, and blood glucose concentrations were analyzed using an IDEXX VetTest chemistry analyzer (IDEXX, Westbrook, ME, USA). The rectal temperature was measured using a veterinary digital thermometer (Vet thermometer^®^, Shang Nong, Qingdao, Shandong, China). The heart and respiratory rates were measured by auscultation using a stethoscope and by observing the chest movements of the patients. The peripheral pulse oxygen saturation (SpO_2_) was monitored by a Masimo Rad-5^®^ pulse oximeter (Masimo, Irvine, CA, USA). Rectal temperature (°C), heart rate (beats/min), respiratory rate (breaths/min), and pulse oxygen saturation (%) were measured every five minutes after successful intubation until the isoflurane was turned off. The average values for the physiological parameters of each participant were used in the data analysis.

Serows were subsequently relocated to the enclosure for recovery with an endotracheal tube secured during transportation and placed in sternal recumbency with 20–40 mL/kg/min of oxygen supplied via the endotracheal tube. The recovery time was defined as the time between the administration of atipamezole and the animal standing. Each individual was observed until walking steadily. The quality of recovery was assessed using the criteria listed in Table 1, which were evaluated and scored by two blinded observers at different time points. The quality of the recovery scoring system was revised in a previous study [27].

### Statistical Analysis

The average doses of the drugs, including tiletamine/zolazepam, ketamine, dexmedetomidine, and atipamezole, were calculated. Boxplots with individual data points were used for the visual assessment of potential outliers and have been presented. The results were reported as mean ± SD or median (if an outlier was noted). The paired Student’s *t*-test or Wilcoxon signed-rank test (if an outlier was noted) was used to compare the anesthetic and physiological parameters of both DZ and DK groups. *p* < 0.05 denoted statistical significance.

## 3. Results

Five Formosan serows (2 males and 3 females) were included in this study. Their ages ranged from 6.1 to 13.9 years old, with an average of 9.7 ± 3.1 years old. The average body weight was 21.8 ± 2.1 kg (range: 19.2 and 26.4 kg). The dosages of dexmedetomidine, tiletamine–zolazepam, and atipamezole used for the DZ group were 22.6 ± 8.3 µg/kg, 2.1 ± 0.25 mg/kg, and 185.6 ± 123.6 µg/kg, respectively. In the DK group, the dosages of dexmedetomidine, ketamine, and atipamezole were 35.8 ± 2.5 µg/kg, 3.6 ± 0.3 mg/kg, and 357.8 ± 25.2 µg/kg, respectively.

The median induction time was 8 min for both groups (Figure 1). All participants were anesthetized within 10 min, except for one animal in the DK group where induction time was 22 min. The median recovery duration (1 min) was the same for the DZ and DK groups, with an outlier recovery duration of 41 min in the DK group (Figure 2). Nonetheless, the recovery score (Figure 3) for the DK group (3.8 ± 0.84) was significantly better than that of the DZ group (2.4 ± 0.55) (*p* = 0.025). Although both combinations induced anesthesia, most serows showed lateral recumbency within 10 min, except for the one with a 22-minute induction time. Furthermore, two out of the five (40%) serows in the DK group showed struggling during the induction phase, paddling, and obvious muscle rigidity after lateral recumbency and could not be intubated successfully on the first attempt such that they then required isoflurane to be delivered via a mask to facilitate the intubation. However, all the serows in the DZ group showed smooth induction without muscle rigidity and could be successfully intubated on their first attempt at the Animal Medical Center.

The physiological parameters are listed in Table 2. Significant differences were observed in the rectal temperature (*p* = 0.008) and respiratory rate (*p* = 0.016), both of which were higher for the DK than for the DZ group. Serows in the DK group demonstrated faster and shallower abdominal respiration patterns than those in the DZ group; however, no major complications were observed throughout the study, except for one serow in the DK group that showed severe hypoxemia during the procedure.

## 4. Discussion

The doses of dexmedetomidine and tiletamine/zolazepam used in this study were within the previously reported dosages for other ruminants [3,28,29]. The qualities of induction with DZ were comparable to that of DK, a published anesthetic combination [3], for Formosan serows.

In this study, dexmedetomidine combined with tiletamine–zolazepam provided satisfactory induction of anesthesia. The serows showed lateral recumbency and were relocated for intubation within 10 min. Additionally, those anesthetized with the DZ combination had fewer complications, including tachypnea and hyperthermia, than those anesthetized with DK. The normal body temperature for Formosan serows has not been published, but it is believed to be similar to that of a goat, which is 38.5–39.7 °C according to the online Merck veterinary manual (https://www.merckvetmanual.com/multimedia/table/normal-rectal-temperature-ranges accessed on 2 May 2024). Thus, the rectal temperature for the DK group in this study was higher than the reference range, which suggested hyperthermia. This change was most likely due to the contribution of zolazepam to the DZ group. Zolazepam is a benzodiazepine with a strong muscle relaxant effect that inhibits the GABA receptors in the spinal cord; therefore, the DZ combination could produce better muscle relaxation and less muscle rigidity, which was reflected by the lower rectal temperature for the DZ group than for the DK group. Ketamine is often combined with benzodiazepines, such as diazepam or midazolam, to reduce undesirable seizure-like activity and muscle rigidity [30]. However, the addition of benzodiazepines to ketamine-based combinations increases the injection volume, which is a significant concern when the combination is to be administered via a blow dart. An increase in the injection volume of the blow dart increases the risk of drug delivery failure and injury to captive or free-ranging wild species [31].

Serows that were administered the DZ combination showed a significantly lower respiratory rate than those administered the DK combination. There was no significant difference between the durations of induction for the groups, but the quality of induction was clinically smoother and characterized by less delirium and struggle during the induction phase for the DZ than for the DK group. It has been reported that the quality of induction can be scored based on the clinical performance of the participant, such as signs of excitation, vocalization, and muscle relaxation, among others [32]. The serows in the DK group showed fast and shallow abdominal respiration patterns, potentially reflecting a poorer quality of induction compared to the DZ group. However, the concentration of the end-tidal carbon dioxide (Et CO_2_) was not recorded in this study because of monitoring limitations. Et CO_2_ should be recorded in future studies to determine the ventilation status after the different anesthetic protocols.

The peripheral oxygen saturation (SpO_2_) was also measured. No significant differences were detected, but the SpO_2_ of the DZ group was higher than that of the DK group. The adverse effects of alpha-2 adrenergic receptor agonists in ruminants include an increase in venous admixture, pulmonary edema, and life-threatening hypoxemia [14,33,34]. Goats show stronger mediated cardiovascular and pulmonary effects of alpha-2 adrenergic receptors than sheep [35], indicating that goats may demonstrate a more marked decrease in cardiac output, increase in systemic vascular resistance, and decrease in oxygen saturation after the administration of potent alpha-2 adrenergic receptor agonists, such as dexmedetomidine. However, profound hypoxemia was not observed in most participants in the present study. Only one serow showed severe hypoxemia (SpO_2_ < 70%) throughout the study. This finding is consistent with the conclusion of a review article that indicated that the hypoxemic response in sheep and goats appears to be individual and probably breed-dependent [36].

The only serow with severe hypoxemia (SpO_2_ < 70%) was found in the DK group, which was due to rumen tympany, one of the most common complications in ruminants during general anesthesia [37]. The diaphragm moves cranially and ventrally when intra-abdominal pressure increases, resulting in a decrease in functional residual capacity and severe hypoventilation, accompanied by hypercapnia and hypoxemia. Placing the animal in sternal recumbency immediately after anesthesia helps eliminate the accumulated gas in the rumen, supplementing the high flow of pure oxygen which thus prevents hypoxemia [38]. Fortunately, the patient did not experience any further complications after emergency treatment, including abdominocentesis or early termination of the procedure. Proper fasting before anesthesia is vital for decreasing the risk of tympanic, regurgitation, and aspiration pneumonia in ruminants. Preanesthetic fasting for 12–24 h is recommended to reduce the volume of rumen content, thereby decreasing the risk of perioperative ruminal tympany and regurgitation [39]. In this study, food and water were withheld from all serows for at least 12 h. No regurgitation was observed during the procedures with this fasting protocol, and the blood glucose concentrations of all the participants were within normal limits.

The quality of recovery was significantly poorer for the DZ group than for the DK group. Unsatisfactory recovery after the administration of tiletamine–zolazepam has been reported in carnivores; clinical signs include prolonged recovery, muscle tremors, and whining [21]. Pharmacokinetically, tiletamine–zolazepam shows a different metabolism in dogs and cats, resulting in different undesired clinical signs during recovery [21]. Furthermore, zolazepam has been reported to metabolize slower than tiletamine in pigs, leading to prolonged recovery after the administration of tiletamine–zolazepam combinations [40]. Merwin et al. found that using tiletamine–zolazepam combined with xylazine hydrochloride prolonged recovery in free-ranging rocky mountain bighorn sheep after the antagonism of xylazine [28]. The findings from Merwin’s study were similar to those of the present study, in which poor quality of recovery was characterized by prolonged recovery, paddling, and ataxia after the administration of atipamezole. One potential reason for this would be the residual effects of tiletamine–zolazepam after dexmedetomidine antagonism.

The quality of recovery after tiletamine–zolazepam anesthesia has been reported to be dose-dependent [5,29]. The average dose of tiletamine–zolazepam used in this study was 2.1 ± 0.25 mg/kg, which was higher than the dose used for anesthesia of wood bison (1.5–3 mg/kg) but within the recommended range based on the study for wood bison [5]. However, the quality of recovery can be improved using a balanced anesthesia technique to reduce the dosage of tiletamine–zolazepam. It is well-known that dogs metabolize zolazepam faster than cats, and the administration of flumazenil does not improve the quality of recovery in tiletamine–zolazepam-anesthetized dogs [21]. Interestingly, it has been reported that the duration of recumbency is unaffected after the administration of flumazenil in llamas with tiletamine–zolazepam anesthesia, indicating that the duration of action is more likely to be influenced by tiletamine than by zolazepam [41]. However, there are currently no pharmacokinetic profiles of tiletamine–zolazepam in Formosan serows. It is difficult to determine whether the administration of flumazenil improves the quality of recovery in Formosan serows. Hence, the administration of flumazenil, an antidote to benzodiazepine, is a potential option for improving the quality of recovery after tiletamine–zolazepam anesthesia in Formosan serows. The pharmacokinetics of tiletamine–zolazepam in Formosan serows are worth investigating in the future.

Alkaline phosphatases comprise a heterogeneous group of enzymes widely distributed in mammalian cells. Alkaline phosphatases have mainly been used to diagnose hepatic diseases, but they have also been reported as useful indicators for diagnosing bone production and diseases, endocrine diseases, genetic and breed-related diseases, and neoplasia [42]. Furthermore, alkaline phosphatase has been used to assess capture myopathy in wild boars [43], mountain goats [44], and free-ranging mosquitoes [45], indicating that alkaline phosphatase may be an indicator of muscular injuries in zoo or wildlife animals. In this study, we attempted to measure the serum concentrations of alanine transaminase, aspartate transaminase, alkaline phosphatase, creatinine kinase, and lactate dehydrogenase by analyzing blood samples from anesthetized participants; however, only the results of alkaline phosphatase were consistently available because of the limitations of the blood analyzer used in the Zoo. Therefore, we decided to use serum alkaline phosphatase concentrations to assess the risk of myopathy due to capture or pursuit and dissociative anesthetic-induced muscular injuries in serows. No significant differences were observed between the groups, and alkaline phosphatase values did not show obvious changes when compared to previous blood work results in the same participant. There are no published serum biochemical values for Formosan serows, but the values in this study were comparable to published serum biochemical values for Southern Chamois, which indicated a reference range of 101–893 U/L for serum alkaline phosphatase [46]. Nevertheless, we could not conclude that the serows were safe from myopathies, because it has been reported that serum alkaline phosphatase may not be specific enough to assess myopathies associated with capture or dissociative anesthetic-induced muscular injuries [43]. Specific indicators, such as alanine transaminase, aspartate transaminase, creatinine kinase, and lactate dehydrogenase, should be monitored and used to assess potential myopathies after anesthesia in Formosan serows.

The blood glucose concentrations have been measured as an indicator of short-term stress response in captive-reared guanacos [24]. The concentrations of blood glucose did not show significant differences between groups in this study; however, an elevation in the blood glucose concentration was observed relative to the previously reported serum biochemistry values of southern chamois [42] and caprine (chemistry reference intervals reported by Animal Health Diagnostic Center of College of Veterinary Medicine, Cornell University). The blood glucose concentrations were >160 mg/dL for both groups in this study, indicating potential short-term stress after anesthesia in Formosan serows, regardless of the protocols used. Elevated blood glucose concentrations have been recognized as indicators of stress in most mammals, and the measurement of serum cortisol concentrations is a more reliable and specific indicator of stress in free-living vertebrates [25]. Nonetheless, Alpha-2 adrenergic receptor agonists have been found to inhibit insulin release, stimulate glucagon release, or both from α and β cells, leading to hyperglycemia [47]. Therefore, blood glucose concentrations may help detect short-term stress. However, monitoring serum cortisol concentrations facilitates stress detection in serows.

The administration route of atipamezole was determined by the clinician based on complications, surroundings in the recovery enclosure, and the temperament of the patient. Several serows experienced arousal within two minutes of intravenous administration of atipamezole, resulting in unsatisfactory recovery. Atipamezole is active following the intramuscular injection, and intravenous administration should be used with caution unless safe recovery has been established and the temperament of the patient is steady [29].

This study had several limitations. First, an anesthetic induction score was not established because the observer was not blinded to the combination during induction; however, the observers for induction and recovery were different, and they were blinded to the combination. Therefore, only the induction time and induction status were recorded, and recovery was scored. Although the two blind overserves were well-trained before scoring the quality of recovery, the level of concordance was not analyzed in this study which might contribute a potential confounding factor to the results. Second, the end-tidal concentration of carbon dioxide was not routinely recorded due to the limitations of the monitor. Third, arterial oxygen tension was not measured because of the unavailability of a blood gas analyzer; however, SpO_2_ was monitored throughout the study and oxygen was supplemented via a patent endotracheal tube. Fourth, the study design was non-randomized due to drug availability. All study participants received DK during the first period and DZ during the second period. Confounding may have occurred if there were differences or changes between the two periods, which was a concern with the prolonged washout between the two periods. This non-randomization also prevented the blinding of the assessor of study outcomes, which may have resulted in information bias, especially for subjective outcome measures such as recovery scores. Furthermore, this study recruited five Formosan serows for a total nine anesthetic protocols in six years (2009–2015); the climatic differences, summer, or winter, could be a factor to influence the quality of the anesthesia and induced different responses to the drug protocols [48]. Nonetheless, the variety of the procedures in this study play a role in affecting the quality of induction and recovery due to different levels of pain. Some of the serows underwent enucleation and castration, which are relatively more invasive and painful procedures compared to annual health examinations, which may have impacted the quality of recovery. Finally, only five Formosan serows were included in this study, resulting in a small sample size. Several non-significant results indicated that the study may not have sufficient power to detect meaningful differences in some outcomes, which could be a practical limitation in studying endemic wildlife.

## 5. Conclusions

The dexmedetomidine–tiletamine–zolazepam combination induced smooth and rapid anesthesia in captive Formosan serows without major complication during maintenance with isoflurane anesthesia when compared to the ketamine–dexmedetomidine combination. However, prolonged and unsatisfactory recovery may be observed in dexmedetomidine–tiletamine–zolazepam-anesthetized Formosan serows after the administration of atipamezole.

## Figures and Tables

**Figure 1 animals-14-01413-f001:**
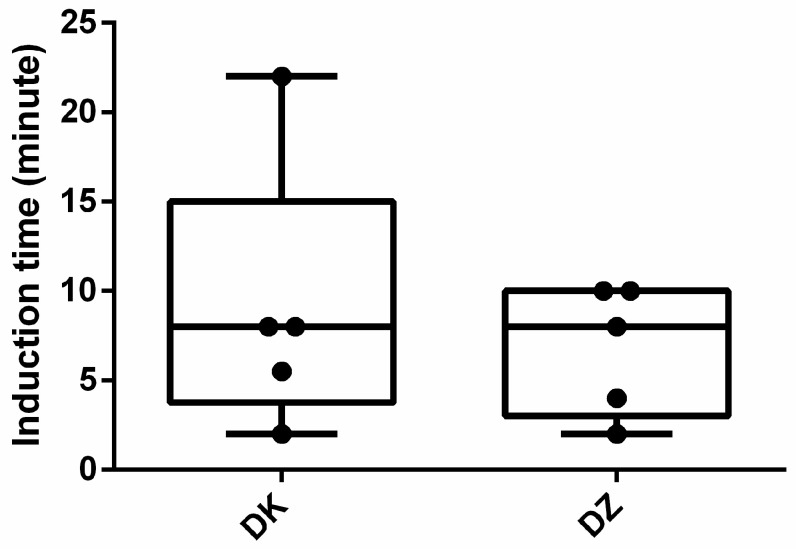
Boxplots of induction durations for the dexmedetomidine–ketamine (DK) and dexmedetomidine–tiletamine–zolazepam (DZ) groups. Dots are individual measures, and the inner horizontal lines are medians.

**Figure 2 animals-14-01413-f002:**
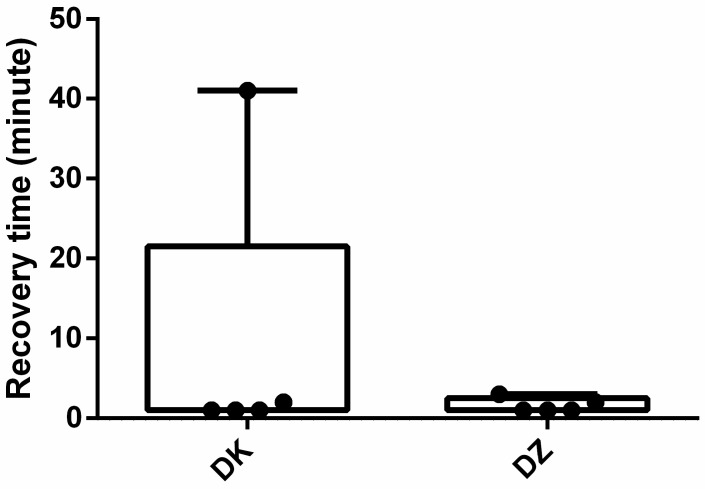
Boxplots of recovery durations for the dexmedetomidine–ketamine (DK) and dexmedetomidine–tiletamine–zolazepam (DZ) groups. Dots are individual measures, and the inner horizontal lines are medians.

**Figure 3 animals-14-01413-f003:**
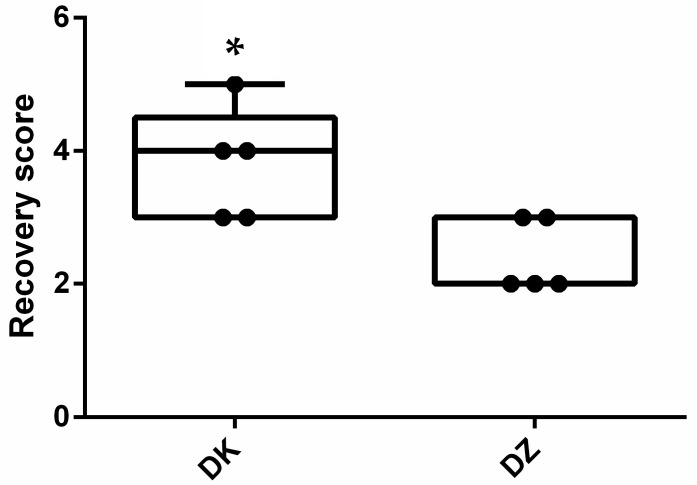
Boxplots of the recovery scores for the dexmedetomidine–ketamine (DK) and dexmedetomidine–tiletamine–zolazepam (DZ) groups. Dots are individual measures, and the inner horizontal lines are medians. * indicates a significant difference between the DK and DZ groups (*p* = 0.025).

**Table 1 animals-14-01413-t001:** Quality of recovery scoring system.

Score	Description
1	Rough recovery, uncoordinated standing with more than five attempts, severe ataxia after standing, and prolonged recovery after administration of atipamezole.
2	Moderately rough recovery, coordinated standing with less than four attempts, moderate ataxia, and prolonged recovery after administration of atipamezole.
3	Relatively calm recovery, coordinated standing with less than three attempts, slight ataxia, and rapid recovery after administration of atipamezole.
4	Calm recovery, coordinated standing with one or two attempts, minor or no ataxia, and rapid recovery after administration of atipamezole.
5	Smooth recovery, coordinated standing with the first attempt and without ataxia, and recovery immediately after administration of atipamezole.

**Table 2 animals-14-01413-t002:** The average values of the physiological parameters during general anesthesia with dexmedetomidine–tiletamine–zolazepam (DZ) and dexmedetomidine–ketamine (DK).

	DZ	DK
Body temperature (°C) *	37.4 ± 0.98	39.91 ± 1.45
Heart rate (beats/minute)	77.4 ± 17.85	104.13 ± 23.88
Respiratory rate (breaths/minute) *	5 ± 1.41	20.25 ± 12.44
SpO_2_ (%)	99 ± 0.71	88.86 ± 9.42
Alkaline phosphatase (U/L)	147.4 ± 39.48	237 ± 94.35
Blood glucose (mg/dL)	161.2 ± 29.02	168.2 ± 45.38

* indicates a significant difference between groups (*p* < 0.05).

## Data Availability

The raw data supporting the conclusions of this study will be made available by the authors upon request.

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
