# Peer review of "Retrospective Comparison of the Anesthetic Effects of Tiletamine–Zolazepam with Dexmedetomidine and Ketamine with Dexmedetomidine in Captive Formosan Serow (Capricornis swinhoei)"

_animals, 2024, doi:10.3390/ani14101413_

Round 1
Reviewer 1 Report
Comments and Suggestions for Authors
A good contribution to a sparce area of knowledge of particular interest to wildlife veterinary medicine. Although, as you indicated there are some major limitations in the design, the data collected is of practical use for future research and anesthetic practice.
You do indicate the major limitations, but as the major variations between the treatments appear to be due to one particular individual case, caution in making the conclusions should be considered. Could the variation be due to the variable effect of the drug administered via blow-dart? Was there any particular aspects of the "outlying" case? The site where the dart lodged (as an injection into fat/ fascia would have a very different uptake to into muscle. Is there any correlations with the length of the procedures (it would be helpful to include the lengths of the total procedures.
25: Re-word - recovery?
100-101: High LDH, CK, AST and ALKP are not causes of capture myopathy, they are results of muscle fiber breakdown. The references do not support this statement.
131-135: Indicate what was the criteria for deciding the actual dosage. Was the visual assessment of body weight the only one done - it seems a bit too precise.
144-145 Ambiguously written - rephrase.
173-174: Indicate the timing of the atipamezole injection.
229-230: Unclear exactly what you are saying.
323: Search further for the application of Alkaline Phosphatases in assessing muscle damage - it is very widely done (and researched).
361: Have you considered that the higher blood glucose could be as result of the 12 hour fasting?
Comments on the Quality of English Language
Generally good. Some expressions and phrases to be improved.
Author Response
Dear reviewer,
Thank you for your review and questions.
We have tried our best to answer your question below and corrected everything that you requested accordingly.
Please see the attachment for the point-by-point response.
We appreciate your help.
Q1. You do indicate the major limitations, but as the major variations between the treatments appear to be due to one particular individual case, caution in making the conclusions should be considered. Could the variation be due to the variable effect of the drug administered via blow-dart? Was there any particular aspects of the "outlying" case? The site where the dart lodged (as an injection into fat/ fascia would have a very different uptake to into muscle. Is there any correlations with the length of the procedures (it would be helpful to include the lengths of the total procedures.
Answer from the authors: Thank you for the comments and questions. The administration of anesthetics was performed by two well-trained and experienced veterinarians with 2-inch needles on the dart; therefore, the chance of missing injection was less likely. The one showed delayed induction was due to potential underestimation of bodyweight which had been discussed in the discussion. Unfortunately, we did not analyze the correlations between the duration of anesthesia and the quality of recovery because most of the regular procedures, including healthy checkup, wound debridement, and so on, took about 120 minutes to complete. The procedures took more than 120 minutes were excluded from the study because of poor ASA-PS ( 3) of the subject and the complexity of the procedure might play a confounding factor in the study. The description of the duration of the procedure has been added in the manuscript.
Q2. 25: Re-word - recovery?
Answer from the authors: Thank you for the comments. It has been revised accordingly.
Q3. 100-101: High LDH, CK, AST and ALKP are not causes of capture myopathy, they are results of muscle fiber breakdown. The references do not support this statement.
Answer from the authors: Thank you for the comments. We are sorry for the misinterpretation and it has been revised accordingly.
Q4. 131-135: Indicate what was the criteria for deciding the actual dosage. Was the visual assessment of body weight the only one done - it seems a bit too precise.
Answer from the authors: Thank you for your question. The criterial for determination of actual dosage was the estimated bodyweight, the temperament of the subject, and the loading volume of the blow dart. Therefore, the dosage of a drug might be fluctuating. For example, if a serow showed anxious and stressful in the morning of study, the dosage of dexmedetomidine might be increased but still within the reference range. The description of the criteria for determination of the actual dosage has been added accordingly.
Q5. 144-145 Ambiguously written - rephrase.
Answer from the authors: Thank you for the comments. It has been revised accordingly.
Q6. 173-174: Indicate the timing of the atipamezole injection.
Answer from the authors: Thank you for the comments. The description of timing of atipamezole was stated in line 141-143. “Atipamezole, the antidote for dexmedetomidine, was administered at the same volume as dexmedetomidine after the procedures, and the animals were returned to their enclosures.”
Q7. 229-230: Unclear exactly what you are saying.
Answer from the authors: Thank you for the comments. We are sorry about your confusion. We were trying to describe that DK was a conventional and published anesthesia protocol for Formosa Serow and we found that the quality of induction with DZ were satisfactory and could be compared with DK. The description has been revised accordingly.
Q8. 323: Search further for the application of Alkaline Phosphatases in assessing muscle damage - it is very widely done (and researched).
Answer from the authors: Thank you for the comments. We understood that ALKP had been used extensively to detect muscle injuries in zoo and wildlife animals and had tried to use it as an indicator to diagnose capture myopathy. Unfortunately, we did not find significant and strong clues to use ALKP as a marker to detect myopathy for Formosan Serows.
Q9. 361: Have you considered that the higher blood glucose could be as result of the 12 hour fasting?
Answer from the authors: Thank you for the questions. Hyperglycemia could be a physiological response due to stress. Overnight fasting, pursuing, and relocation for anesthesia could be factors to induce stress for the serow. Therefore, the reasons of hyperglycemia in this study were multifactorial but most likely could be concluded as “stress-induced”. Nonetheless, dexmedetomidine has be reported as a potential factor to increase blood glucose level due to inhibition of insulin release.

Reviewer 2 Report
Comments and Suggestions for Authors
The study is interesting because anesthetic protocols suitable for wild ruminants are necessary. However, I have some doubts.
Materials and methods:
Enter the number of animals used in the study;
Specify how you established the weight of the animals, the actual weight from which the actual dosage of drugs used derives;
Enter the manufacture and trade names of the instruments and drugs used in the study;
Results
Line 180-181 also report in materials and methods
Statistic analysis
he results given the small sample. However, the Wilcoxon test and the Student's t test for paired data are used not to compare two groups but to evaluate changes along the time line. in this study it is not clear whether the physiological parameters were recorded over a period of time. Therefore, since these are non-parametric data, it is appropriate to express them with median and range and make the comparison between the groups using the Mann whitney test.
State whether you have calculated the level of concordance w between the observers who assigned the scores
Discussion
Within the limits of the study, insert that the temperature difference could have an influence on the results, to support this concept you can comment on the following bibliography:
Giovanna Costa, Marcello Musico, Filippo Spadola, Matteo Oliveri, Fabio Leonardi, Claudia Interlandi. (2021). Comparison Of Tiletamine-Zolazepam Combined With Dexmedetomidine Or Xylazine For Chemical Immobilization Of Wild Fallow Deer (Dama Dama). Journal Of Zoo And Wildlife Medicine, p. 1009-1012, ISSN: 1042-7260, doi: 10.1638/2019-0140
2019 Spadola F., Costa G., Interlandi C., Musicò Marcello (2019). Hyaluronidase, With Xylazine And Ketamine, Reducing Immobilization Time In Wild Cattle (Bos Taurus). Large Animals Review, vol. 25, p. 159-161, ISSN: 1124-4593
Comments on the Quality of English Language
Minor editing of English language required
Author Response
Dear reviewer,
Thank you for your review and questions.
We have tried our best to answer your question below and corrected everything that you requested accordingly.
Please see the attachment for the point-by-point response.
We appreciate your help.
Q1. The study is interesting because anesthetic protocols suitable for wild ruminants are necessary. However, I have some doubts.
Materials and methods:
Enter the number of animals used in the study;
Answer from the authors: Thank you for the comments. It has been added accordingly.
Q2. Specify how you established the weight of the animals, the actual weight from which the actual dosage of drugs used derives;
Answer from the authors: Thank you for the comments. The criterial for determination of actual dosage was the estimated bodyweight, the temperament of the subject, and the loading volume of the blow dart. Therefore, the dosage of a drug might be fluctuating. For example, if a serow showed anxious and stressful in the morning of study, the dosage of dexmedetomidine might be increased but still within the reference range. The description of the criteria for determination of the actual dosage has been added accordingly.
Q3. the manufacture and trade names of the instruments and drugs used in the study;
Answer from the authors: Thank you for the comments. It has been added accordingly.
Q4. Results
Line 180-181 also report in materials and methods
Answer from the authors: Thank you for the comments. It has been added accordingly.
Q5. Statistic analysis
he results given the small sample. However, the Wilcoxon test and the Student's t test for paired data are used not to compare two groups but to evaluate changes along the time line. in this study it is not clear whether the physiological parameters were recorded over a period of time. Therefore, since these are non-parametric data, it is appropriate to express them with median and range and make the comparison between the groups using the Mann whitney test.
Answer from the authors: Thank you for the comments. We had discussed with the statistician again regarding the result analysis and found that student’s t-test remained the most accurate way to compare results from different two groups; however, to minimize the impact of outlier, we used Wilcoxon signed-rank test if an outlier was noted to ensure the accuracy of the results.
Q6. State whether you have calculated the level of concordance w between the observers who assigned the scores
Answer from the authors: Thank you for the comments. We did not analyze the concordance between the observers. Both observers were well trained before scoring the quality of recovery. The description about the analysis of concordance has been added accordingly in discussion.
Q7. Discussion
Within the limits of the study, insert that the temperature difference could have an influence on the results, to support this concept you can comment on the following bibliography:
Answer from the authors: Thank you for the value opinion. It has been added accordingly.

Reviewer 3 Report
Comments and Suggestions for Authors
Simple summary and abstract
Line 16: Pay attention to the concept of safe under anaesthesia. I suggest revising the sentence. Many experts in the field speak of a 'safe anaesthetist but never of protocol';
Introduction
Line 56: Same concept as above;
Line 59-61: I do not understand why authors speak about commercial solution. In my opinion is unnecessary, I suggest removing it;
Line 61: Same concept as above;
Line 71: reference please;
Line 74: Mentioned references (10-12) are dated. I does not if the paper was written immediately after cases were recorded. If it is, it is understandable why references are dated. In any case, for type of reference I suggest using more recent references, for example:
Cat: doi: 10.3390/vetsci9100520;
Dog: doi: 10.1111/jsap.13668;
Sheep: doi: 10.1016/j.vaa.2018.06.009;
Goats: doi: 10.1002/vms3.732.
Line 107-114: The aims of the study are not very clear, especially when comparing these lines with the study design. I suggest remodeling the sentence;
Line 124: “Anesthesia was administered as part of the routine 123 health care of the serows for procedures, including annual health examinations, trauma management, castration, enucleation, and ophthalmologic examinations” Animals inclusion criteria are so large that it is difficult to recreate a standard and reproducible sample. The clinical starting conditions of each individual animal influence the depth of sedation. How do you think this is not a potential bias for the study?
Line 131: “The dosage of the anesthetic was calculated based on body weight from visual measurements and previous medical records. For the DZ combination, the reference dose of dexmedetomidine was 40–60 μg/kg, and that of tiletamine-zolazepam was 2.2–4.2 mg/kg. For the DK combination, the reference dose was 30–50 μg/kg for dexmedetomidine and 3–4 mg/kg for ketamine.” How do you think that can not influence the study? Especially for the wide range of dexmedetomidine used.
Line 141: For my knowledge, the administration routes of atipamezole are subcutaneous or intramuscular. Are you sure that is licensed also for intravenous administration? The efficacy of intravenous atipamezole is known but generally is used just in emergency case. If you are agree, it is reasonable specify it;
Line 143: I did not understand whether vital parameter assessments were carried out post-administration of atipamezole;
Line 164: which brand of thermometer?;
Line 180-187: statistics (consequently, the results should be revised). On what basis was the number of subjects chosen? A power study explained in detail should be shown. was a normality study conducted? if so, why was it not presented? A consort diagram is desirable;
Line 191: the weights of the animals are very large to make an even dosage of drug, especially for dexmedetomidine;
Line 218-225: These parameters are taken every 5 minutes but it does not say from when, how long and until when, it is asked to specify;
Results
to be reassessed following the revision of statistics
Discussion
the discussion is too long with arguments outside the scope of the paper
Conclusion
fine
Author Response
Dear reviewer,
Thank you for your review and questions.
We have tried our best to answer your question below and corrected everything that you requested accordingly.
Please see the attachment for the point-by-point response.
We appreciate your help.
Simple summary and abstract
Q1. Line 16: Pay attention to the concept of safe under anaesthesia. I suggest revising the sentence. Many experts in the field speak of a 'safe anaesthetist but never of protocol';
Answer from the authors: Thank you for the comments. It has been revised accordingly.
Q.2 Line 56: Same concept as above;
Answer from the authors: Thank you for the comments. It has been revised accordingly.
Q3. Line 59-61: I do not understand why authors speak about commercial solution. In my opinion is unnecessary, I suggest removing it;
Answer from the authors: Thank you for the comments. The purpose of this description was trying to help more readers worldwide to understand the exact combination of tiletamine – zolazepam has different trade name in different countries. Most of the veterinarians in Asia do not know that TelazolÒ in the United States is identical to ZoletilÒ in Asia, and vice versa.
Q4. Line 61: Same concept as above;
Answer from the authors: Thank you for the comments. The purpose of this description was to indicated that tiletamine-zolazepam showed wider safety margin and better muscle relaxation compared to ketamine.
Q5. Line 71: reference please;
Answer from the authors: Thank you for the comments. It has been added accordingly.
Q6. Line 74: Mentioned references (10-12) are dated. I does not if the paper was written immediately after cases were recorded. If it is, it is understandable why references are dated. In any case, for type of reference I suggest using more recent references, for example:
Answer from the authors: Thank you for the value comments. The manuscript was written after the study
and we were trying to use the most original studies to introduce the concept of this study. Thank you again
for your input, but we decided to keep using the most original articles for introduction.
Q7. Line 107-114: The aims of the study are not very clear, especially when comparing these lines with the study design. I suggest remodeling the sentence;
Answer from the authors: Thank you for the comments. The description has been revised accordingly.
Q8. Line 124: “Anesthesia was administered as part of the routine 123 health care of the serows for procedures, including annual health examinations, trauma management, castration, enucleation, and ophthalmologic examinations” Animals inclusion criteria are so large that it is difficult to recreate a standard and reproducible sample. The clinical starting conditions of each individual animal influence the depth of sedation. How do you think this is not a potential bias for the study?
Answer from the authors: Thank you for the question. Although the purposes of anesthesia for the subjects were different, only the subjects with American Association of Anesthesiologist – physical status (ASA-PS) 1 – 2 were included in the study, indicating that the anesthesia associated risks were minimal and the patients were under similar healthy status.
Q9. Line 131: “The dosage of the anesthetic was calculated based on body weight from visual measurements and previous medical records. For the DZ combination, the reference dose of dexmedetomidine was 40–60 μg/kg, and that of tiletamine-zolazepam was 2.2–4.2 mg/kg. For the DK combination, the reference dose was 30–50 μg/kg for dexmedetomidine and 3–4 mg/kg for ketamine.” How do you think that can not influence the study? Especially for the wide range of dexmedetomidine used.
Answer from the authors: Thank you for your question. The criterial for determination of actual dosage was the estimated bodyweight, the temperament of the subject, and the loading volume of the blow dart. Therefore, the dosage of a drug might be fluctuating. For example, if a serow showed anxious and stressful in the morning of study, the dosage of dexmedetomidine might be increased but still within the reference range. The description of the criteria for determination of the actual dosage has been added accordingly. Certainly, the dose of dexmedetomidine might be an important factor to determine the duration of onset of the combination; however, based on the result of this study, all subjects could be induced within 10 minutes except one in the DK group. We would like to explore the impacts of dexmedetomidine on the duration of onset of induction in the future if we can recruit more Formosa serows in the study.
Q10. Line 141: For my knowledge, the administration routes of atipamezole are subcutaneous or intramuscular. Are you sure that is licensed also for intravenous administration? The efficacy of intravenous atipamezole is known but generally is used just in emergency case. If you are agree, it is reasonable specify it;
Answer from the authors: Thank you for your comments. Yes, we agree with your value opinion. We administered atipamezole intravenously only for subjects experienced intraoperative complications; others were receiving atipamezole half IV and half IM which was a recommended administration route for zoo animals.
Q11. Line 143: I did not understand whether vital parameter assessments were carried out post-administration of atipamezole;
Answer from the authors: Thank you for your question. The physiological parameters were only during anesthesia. The only parameter that we recorded after administration of atipamezole was the quality of recovery.
Q12. Line 164: which brand of thermometer?;
Answer from the authors: Thank you for your question. The brand and manufacture has been added accordingly.
Q13. Line 180-187: statistics (consequently, the results should be revised). On what basis was the number of subjects chosen? A power study explained in detail should be shown. was a normality study conducted? if so, why was it not presented? A consort diagram is desirable;
Answer from the authors: Thank you for your question. The number of study subjects was determined by the maximum numbers of Formosan serows that we could recruit in the study, resulting one of the limitations of the study – small sample size and potential non-sufficient power to detect differences in some outcomes. The normality test was performed, and normal distribution was not found in several parameters because of several outliers; therefore, Wilcoxon signed-rank test was used if an outlier was noted to ensure the accuracy of the interpretation of the results. Boxplots with individual data points were used for the visual assessment of potential outliers and have been presented.
Q14. Line 191: the weights of the animals are very large to make an even dosage of drug, especially for dexmedetomidine;
Answer from the authors: Thank you for your comments. Although the body weight varies between individuals, we calculated every drug by mcg/kg or mg/kg to ensure the precision of the dosage calculation.
Q15. Line 218-225: These parameters are taken every 5 minutes but it does not say from when, how long and until when, it is asked to specify;
Answer from the authors: Thank you for your comments. The measurement was begun after successful intubation until isoflurane was turned off. The detailed description has been added accordingly.
Results
Q15. to be reassessed following the revision of statistics
Answer from the authors: Thank you for your comments. The statistician recommended that we would keep the original statics for your review.
Discussion
Q16. the discussion is too long with arguments outside the scope of the paper
Answer from the authors: Thank you for your comments. We were trying our best to cover the whole aspects of this study and discussing the results point-by-point. We do appreciate your value opinion, but we would like to leave the final decision to the editor to see if we need to shorten the discussion.

Round 2
Reviewer 3 Report
Comments and Suggestions for Authors
Dear authors,
Thank you for your comments.
Congratulations